# Impact of a Dedicated Pretransplant Infectious Disease Consultation on Respiratory Tract Infections in Kidney Allograft Recipients: A Retrospective Study of 516 Recipients

**DOI:** 10.3390/pathogens12010074

**Published:** 2023-01-03

**Authors:** Elsa Feredj, Etienne Audureau, Anna Boueilh, Vincent Fihman, Slim Fourati, Jean-Daniel Lelièvre, Sébastien Gallien, Philippe Grimbert, Marie Matignon, Giovanna Melica

**Affiliations:** 1Infectious Disease Department, Groupe Hospitalier Henri-Mondor/Albert Chenevier, Assistance Publique-Hôpitaux de Paris (AP-HP), 94010 Créteil, France; 2IMRB (Institut Mondor de Recherche Biomédicale), Université Paris-Est-Créteil (UPEC), INSERM U955, Equipe 16, 94010 Créteil, France; 3Department of Public Health, Hôpitaux Universitaires Henri Mondor, Assistance Publique—Hôpitaux de Paris, 94010 Créteil, France; 4Nephrology and Renal Transplantation Department, Groupe Hospitalier Henri-Mondor/Albert-Chenevier, Assistance Publique-Hôpitaux de Paris (AP-HP), 94010 Créteil, France; 5Virology, Bacteriology and Infection Control Units, Clinical Microbiology Department, AP-HP (Assistance Publique-Hôpitaux de Paris, 94010 Créteil, France; 6IMRB (Institut Mondor de Recherche Biomédicale), VIC (Virologie Immunité Cancer), DHU (Département Hospitalo-Universitaire), Université Paris-Est-Créteil (UPEC), INSERM U955, Equipe 18, 94010 Créteil, France; 7Ecole Vétérinaire de Maison Alfort, EA Dynamyc, Université Paris Est Créteil, 94000 Créteil, France; 8IMRB (Institut Mondor de Recherche Biomédicale), VIC (Virus-Immunité-Cancer), DHU (Département Hospitalo-Universitaire), Université Paris-Est-Créteil (UPEC), INSERM U955, Equipe 21, 94010 Créteil, France; 9Clinical Investigation Center-Biotherapies 504, Groupe Hospitalier Henri-Mondor/Albert Chenevier Assistance Publique-Hôpitaux de Paris (AP-HP), 94010 Créteil, France

**Keywords:** vaccination, pneumonia, kidney transplantation

## Abstract

Background: Respiratory tract infections (RTIs) are a leading cause of death after kidney transplant. Preventive strategies may be implemented during a dedicated infectious disease consultation (IDC) before transplantation. Impact of IDC on RTIs after transplant has not been determined. Methods: We conducted a monocentric retrospective cohort analysis including all kidney transplant recipients from January 2015 to December 2019. We evaluated the impact of IDC on RTIs and identified risk and protective factors associated with RTIs. Results: We included 516 kidney transplant recipients. Among these, 145 had an IDC before transplant. Ninety-five patients presented 123 RTIs, including 75 (61%) with pneumonia. Patient that benefited from IDC presented significantly less RTIs (*p* = 0.049). RTIs were an independent risk factor of mortality (HR = 3.64 (1.97–6.73)). Independent risk factors for RTIs included HIV (OR = 3.33 (1.43–7.74)) and HCV (OR = 3.76 (1.58–8.96)). IDC was identified as an independent protective factor (OR = 0.48 (0.26–0.88)). IDC prior to transplantation is associated with diminished RTIs and is an independent protective factor. RTIs after kidney transplant are an independent risk factor of death. Implementing systematic IDC may have an important impact on reducing RTIs and related morbidity and mortality.

## 1. Introduction

Over 80,000 patients worldwide benefit from kidney transplantation every year [1]. The immunosuppressive treatment required to prevent organ rejection places the transplant recipient at an increased risk for community-acquired, nosocomial and donor-derived infections, as well as reactivation of latent infections [2]. The spectrum of implicated pathogens depends on the host’s net state of immunosuppression, his/her past exposure history, and pathogen-dependent factors [3]. Despite the important decrease in transplantation-related morbidity and mortality and generalization of infection preventive strategies, infection remains one of the leading causes of death after kidney transplant [4,5,6]. Indeed, improvement in transplantation management may be counterbalanced by less stringent transplantation eligibility criteria of kidney recipients and donors, such as age or existing co-morbidities [7,8,9].

Among infections, bacterial urinary and respiratory tract infections are the most frequent after kidney transplantation [4,10] It is estimated that up to 20% of kidney allograft recipients will suffer from respiratory tract infections (RTIs) [11,12]. RTIs develop regardless of post-transplant period and immunosuppressive regimen [13], and kidney transplant patients are particularly susceptible to acute respiratory distress syndrome (ARDS) and respiratory failure upon infection [14]. The occurrence of RTIs adversely affects graft survival, morbidity, and mortality after kidney transplantation [15,16,17]. Therefore, preventive strategies have been established, including screening for and treatment of latent tuberculosis, prophylaxis against *Pneumocystis jirovecii*, and systematic update of the recommended vaccine schedule before transplantation. These strategies may be implemented during a dedicated infectious disease consultation (IDC) [18,19], resulting in better adherence to vaccination before transplantation [20], and multiple transplantation centers have integrated IDC in common practice [21].

Despite these preventive efforts, the impact of an IDC on RTI occurrence, as well as an extensive epidemiological characterization of RTIs after kidney transplantation are still poorly documented. To address these questions, we conducted a retrospective analysis of all kidney transplant recipients over 5 years in our center. We analyzed the association between IDC before transplantation and the subsequent occurrence of post-transplantation RTI. We also characterized the spectrum, the incidence, and the risk factors of RTI and their association with morbidity and mortality.

## 2. Patients and Methods

### 2.1. Patient Population and Study Design

We conducted a retrospective cohort study including all kidney allograft recipients engrafted from January 2015 to December 2019 in Henri Mondor university hospital. Demographic, clinical, laboratory, treatments, and outcomes data of recipients and donors were collected from electronic medical records using a standardized data collection form. Transplant recipient’s cohort description was authorized by the local ethic committee (IRB #00003835) and is in accordance with the Helsinki Declaration of the World Medical Association. The patients’ written consent was obtained.

RTIs were considered according to the current literature and international guidelines [22,23]. We consider both bronchitis and pneumonia: bronchitis is defined by clinical criteria of an acute pulmonary infection (cough, dyspnea, sputum, and fever) without compatible radiological image nor auscultatory abnormalities; pneumonia is defined as an association of clinical criteria of acute pulmonary infection and compatible radiological image on chest X-ray or CT scan and/or auscultation abnormalities.

All recipients were followed at least one year after transplantation or until death or allograft loss. Glomerular filtration rate (eGFR) was estimated using the Modification of Diet in Renal Disease (MDRD) formula [24]. Expanded criteria donor (ECD) was defined as donors older than 60 years or between 50 and 60 years with at least two of the following criteria: hypertension, preretrieval serum creatinine > 130 µmol/L and cerebrovascular cause of death [25].

### 2.2. Infectious Disease Consultation before Kidney Transplantation

IDC was progressively implemented starting January 2014 in our center for patients awaiting kidney transplantation, regardless of medical history and comorbidities. It is performed regardless of other indications that also require an infectious disease expertise such as HIV or HCV infection. The number of IDCs increased over time as new patients were listed on the French national registry. The aim was to update recommended vaccination schedules based on the French national vaccination recommendations for patients awaiting solid organ transplantation [26] and to treat latent tuberculosis. Information regarding types and timing of previous vaccinations against hepatitis A and B, pneumococcus, and influenza, tetanus–diphtheria–polio–acellular pertussis (TDPaP) was recorded. Serological status of hepatitis A and B, varicella-zoster virus (VZV), measles, and yellow fever in previously vaccinated patients was evaluated. The catch-up strategy for vaccinations based on the French national vaccines schedule and recommendations for immunocompromised patients/patients awaiting solid organ transplantation was proposed and personalized as follow:All patients: flu and pneumococcus vaccinations;Only if a recall dose needed: tetanus–diphtheria–polio–acellular pertussis (TDPaP) vaccination;Older than 65 years without previous immunosuppression: attenuated VZV vaccine;Younger than 19 years, female or men having sex with men (MSM): papillomavirus vaccine;On demand if needed (travel or catch-up doses): measles, mumps, rubella (MMR), VZV, meningococcus, haemophilus influenza, yellow fever hepatitis, and A and hepatitis B vaccinations. Yellow fever is mandatory for travelers to endemic intertropical regions of Africa and South America.

Latent tuberculosis treatment was recommended in case of positive interferon gamma release assay (IGRA) and/or radiological evidence of old and healed tuberculosis lesions, not previously treated for tuberculosis [27]. Treatment included isoniazid for 6 months. Individual infectious risk factors were assessed and prophylaxis recommendations and safe living strategies after transplant were suggested. In case of dialysis, vaccinations were performed and updated in the dialysis center, after the consultation. Additional prophylaxis included: CMV prophylaxis that involved the administration of oral valganciclovir to high (D+/R−) and intermediate (R+ treated with thymoglobulin) risk patients. Duration of prophylaxis was 6 months in high-risk patients and 3 months in intermediate ones. *Pneumocystis jirovecii* prophylaxis included trimethoprim–sulfamethoxazole (400 mg per day) or pentacarinat aerosol monthly for 12 months after transplantation and until CD4 count was above 200/mm^3^.

Clinical (past personal history), biological, and radiological evaluation was performed to evaluate individual infectious, and post-transplant prophylaxis was adapted following recommendations.

### 2.3. Endpoints

The primary endpoint was the occurrence of RTI (bronchitis and pneumonia), so as to evaluate a potential protective effect of pretransplant IDC and identify key risk factors. Secondary endpoints included death and allograft loss to evaluate the potential influence of RTIs on morbimortality, independently of other influent prognostic factors.

### 2.4. Statistical Analysis

Continuous variables are presented as mean (±Standard Deviation (SD)) or median (Interquartile Range (IQR)). Categorical variables are presented as counts (%). Baseline donor, recipient, and kidney transplant characteristics were compared between patients with and without RTIs using t-tests, or Wilcoxon–Mann–Whitney tests for continuous variables, and Chi-2 or Fisher’s exact tests for categorical variables, as appropriate.

To assess risk/protective factors of RTIs, we first conducted logistic regression models, calculating odds ratios along with their 95% confidence intervals (CI95%). The association between pretransplant IDC and subsequent occurrence of RTIs was further assessed in a time-to-event framework, using Fine–Gray regression modeling accounting for the competing risk of death, and calculated the resulting subhazard ratio (SHR) along with its CI95%; the association was illustrated by plotting the corresponding cumulative incidence curves.

To assess the potential effect of RTIs on all-cause death, we used time-dependent Cox proportional hazards regression, considering RTIs as a time-varying covariate, and computing hazard ratios (HR) along with their CI95%. Extended Kaplan–Meier survival curves accounting for RTIs as a time-varying covariate were plotted to illustrate the association [28].

Factors entered in multivariate analyses were those found to be associated with the endpoint at *p* < 0.2 in univariate analyses, then applying a stepwise backward selection procedure by sequentially removing variables not significant at *p* < 0.05 until the final model was reached.

No imputation of missing data was conducted. All tests were two-tailed, considering *p*-values < 0.05 as significant. All analyses were performed using Stata v16.1 (Statacorp, College station, TX, USA).

## 3. Results

### 3.1. Study Population

The flow chart of patients included in the study is presented in Figure 1. Between January 2015 and December 2019, 516 kidney transplantations were performed. Baseline characteristics of the study population are presented in Table 1. Median age was 55.6 years (44.1–64.8). Median time from transplantation was 12 (4–30) months. Among the 516 patients, 145 underwent IDC. At the end of the follow-up period, RTI rate was 18.4%, patient mortality 10.5%, and allograft loss 6.4%. Immunosuppressive regimen was decided following local protocol by the nephrologist and depending on rejection risk score and personal history. Type of immunosuppressive treatments, induction, and maintenance, were similar in both groups (Appendix A). Follow-up characteristics of the patients are presented in Table 2.

### 3.2. Risk and Protective Factors of RTIs

In unadjusted logistic regression analysis, risk factors for RTIs after kidney transplantation were recipient age (OR = 1.02 (1.00–1.04), HIV and HCV infection (OR = 4.51 (2.21–9.23) and OR = 6.34 (2.93–13.70), respectively), and combined liver/kidney transplants (OR = 2.65 (1.13–6.19)). In contrast, IDC before transplantation was associated with 50% reduction in RTIs (OR = 0.50 (0.28–0.88)) (Appendix A). Multivariable analysis found HIV, HCV, recipient age, and pretransplant donor-specific anti-HLA antibodies as independent risk factors of RTI, while IDC was confirmed as an independent protective factor (OR = 0.48 (0.26–0.88) (Table 3). Cumulative incidence estimates of RTI according to pretransplant IDC are shown in Figure 2, confirming a protective effect in a competing risk time-to-event analysis framework (SHR = 0.59 [0.35–0.99], *p* = 0.049).

### 3.3. Potential Effect of RTIs on Patient and Allograft Survival

Within follow-up, we recorded 54 deaths (10.5%) and 33 allograft losses (6.4%). While overall allograft loss rate did not significantly differ between both groups (RTIs vs. control group, 4.2% vs. 6.9%, *p* = 0.55), overall survival was significantly lower in the RTIs group (23.2% vs. 7.6% *p* < 0.0001, time-dependent Cox model HR = 3.64 (1.97–6.73) (Figure 3A). Occurrence of RTI was an independent risk factor of mortality after kidney transplantation (HR = 1.93 (1.02–3.70), together with HIV and HCV seropositivity, recipient age, and combined kidney and liver transplantation (Table 4—Model 1). These results are confirmed when considering the potential effect of pneumonias on overall survival (Figure 3B, Table 4—Model 2).

### 3.4. Characteristics of RTIs

A total of 123 RTI episodes in 95 patients were found. Median time from transplantation to RTI episode was 5 (1–12) months. Among RTIs, 48/123 (39%) bronchitis and 75/123 (61%) episodes of pneumonia were reported, of which 48/75 (64%) were documented with 71 pathogens. Among documented pathogens for pneumonia, bacteria were the leading cause of infection, accounting for 38/71 (53.5%) of identified pathogens, followed by viruses (N = 23/71; 32.4%) and fungi (N = 5/71; 7%). Considering bacteria, we observed N = 30/71 (81%) Gram-negative pneumonia, including N = 12 *Pseudomonas aeruginosa* (32.4%), N = 5 *Enterobacter cloacae* (13.5%), N = 3 *Klebsiella pneumoniae* (8.1%), N = 3 *Haemophilus influenzae* (8.1%), N = 3 *Legionella* (8.1%), N = 2 *Escherichia coli* (5.4%), and others (*Acinetobacter haemolyticus* N = 1 and *Moraxella catarrhalis N = 1*). One episode of mycobacterium tuberculosis was also recorded. Among Gram-positive bacteria, we identified five (13.5%) *Pneumococcus*, one *Staphylococcus aureus*, and one *Enterococcus* infection. Viral infections were found in 23/71 (37.3%) pneumonias, including *influenza virus* N = 11 (47.8%) and other respiratory viral pathogens (N = 12). We also identified five cases of invasive fungal infections, accounting for 7% of identified pathogens with three invasive aspergillosis and two cases of *Pneumocystis jirovecii*. Multiple pathogens (two or more) were identified in 10 (8.1%) episodes. Most pneumonia episodes required hospitalization N = 73/75 (97.3%), more than half (52%) in intensive care units (39/75), and 26/75 (35%) required mechanical ventilation. Pneumonia episodes led to death in 9/75 (12%) patients.

## 4. Discussion

By conducting a large retrospective analysis of 516 kidney transplant recipients over 5 years, we demonstrated the benefits of a systematic IDC prior to transplantation in reducing the occurrence of RTIs. We also found that RTIs are an independent risk factor of death after kidney transplantation, with critical disease occurring in half of infected patients. Our results are in line with previous studies that described the high infection burden in kidney transplant patients and its major impact on mortality [4,11,12,15,29,30], as well as highlight the relevant contribution of respiratory tract infections.

Interestingly, a systematic infectious disease consultation before transplantation emerged as a significant protective factor for RTIs. We believe that implementing measures such as the treatment of potential latent tuberculosis infection and the update of the recommended vaccines [19,26,31] prior to transplantation should be an integral part of the pretransplantation process. IDC permits a tailored evaluation of the infectious risk for each patient, before the graft and immunosuppression initiation [20]. Indeed, some vaccine-preventable infections have been associated with significant morbidity and mortality in solid organ transplant recipients [32,33], such as pneumococcal pneumonias, and the efficacy of post-transplantation vaccination is dramatically diminished upon immunosuppressive drugs initiation, with low seroconversion and seroprotection rates after vaccinations [34,35]. Previous studies have shown that IDC improves the management of kidney transplant recipients with resulting higher rates of vaccination and lower rates of hospitalization [21,36]. Indeed, IDC permitted to increase the rates of pneumococci vaccination from 6% to 99% of patients [20]. Unfortunately, the spectrum of respiratory pathogens was not analyzed according to IDC. Nevertheless, and most probably directly related to higher vaccination rates, no pneumococcal infection was documented in the IDC group, and only three influenza infections. Interestingly, the IDC group was also protected from infections not directly covered by measures undertaken during IDC, such as Gram-negative bacteria-related RTIs. It is possible that health and lifestyle advice may have indirect protective effects by limiting hospitalizations and subsequent exposure to nosocomial bacteria such as Gram-negative bacteria. Moreover, vaccine-mediated trained immunity may also provide broader protective effects to other infectious agents [37]. Additionally, this study was performed before the COVID-19 pandemic, which has strongly impacted the landscape of pulmonary infections. The introduction and the efficiency of SARS-CoV-2 vaccines supports systemic IDC prior to transplantation.

To our knowledge, this study is the first to report a clinical impact of IDC on pulmonary infection rates. Contrasting with the impact of RTIs on mortality, we did not find a significant association between IDC and patient survival, most probably due to a relatively short-term follow-up period.

In this study, age and pre-existing infections with HCV or HIV were identified as risk factors associated with RTIs and patient mortality This is not surprising since the last decade has witnessed an extension of renal transplantation with less stringent eligibility criteria [38]. Our study emphasizes the need to individually assess older patients or patients with other immunosuppressive conditions, such as pre-existing chronic infection (HIV and/or HCV), and to adopt personalized preventive strategies in these populations. Pulmonary infections are frequent complications of HIV infection, regardless of the degree of immunosuppression [39], and pneumococcal infections in particular [40]. The impact of HCV infection is less clear but could be related to the underlying chronic liver disease that may persist after HCV cure [41,42,43,44].

By analyzing the spectrum of respiratory pathogens, we found that Gram-negative bacteria were the more frequently documented microbes, in particular *Pseudomonas aeruginosa*, followed in frequency by other *Enterobacteriaceae* species. The predominance of Gram-negative bacteria may be related to predisposing conditions such as long and repeated hospital stays, older age of transplant recipients and colonization and/or subclinical infections in this population [45]. In line with our findings, some studies have reported an evolution towards Gram-negative bacteria’s predominance after solid organ transplant [46,47], as well as a time-dependent increase in the rate of antibiotic resistant species, including *Pseudomonas aeruginosa* [48,49,50]. The spectrum of challenging nosocomial bacteria with increased resistance rates may, in part, explain the severity of respiratory infections in transplant recipients.

Our study has some limitations. First, it is a monocentric retrospective study that included all renal transplant patients in our center. Despite the inclusion of more than 500 patients, our findings require validation from other centers applying similar preventive strategies in a dedicated consultation. Second, the retrospective nature of our study, and the fact that IDC was implemented progressively, may have introduced confusion by indication, where patients with more severe comorbidities were referred to IDC. Nevertheless, IDC was progressively and sequentially implemented regardless of medical history, and in the case where such bias would have been introduced, the observed impact of IDC would be underestimated. Third, the relatively short follow-up period from renal transplant may not be sufficient to identify the impact of IDC on patient survival outcome. Finally, although we have previously shown that IDC increased vaccination adherence [20], our analysis did not include the impact of IDC on vaccination coverage.

Altogether, our study highlighted a protective effect of IDC prior to transplantation on the occurrence of RTIs, identified age, HCV, and HIV as risk factors for RTIs and mortality, and provided a detailed characterization of the spectrum of respiratory tract infections after renal transplantation. In light of our results, we highly recommend that all patients eligible for renal transplantation attend a dedicated infectious disease consultation to implement personalized preventive strategies prior to transplantation.

## Figures and Tables

**Figure 1 pathogens-12-00074-f001:**
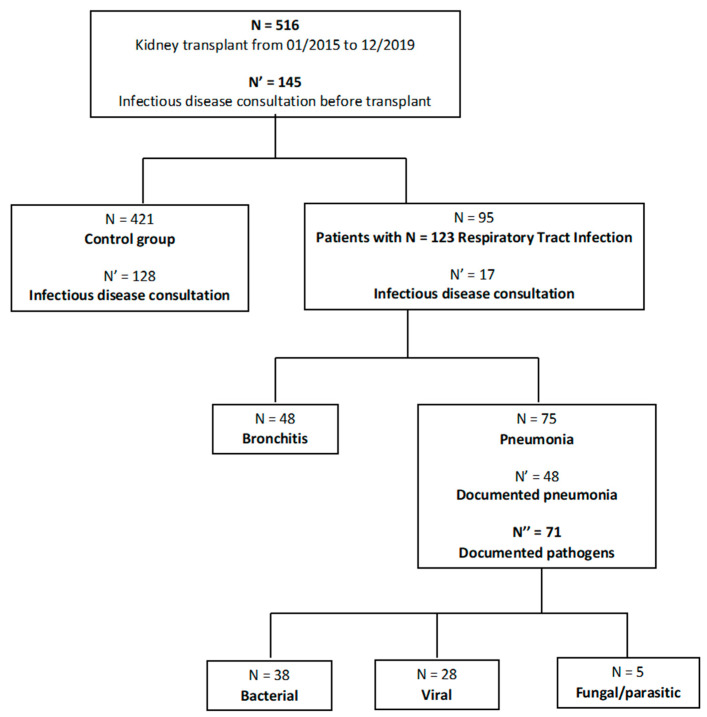
Flow chart of the study population. Between January 2015 and December 2019, 516 kidney transplantations were performed in *n* = 516 patients. Among them, 145 patients benefited from infectious disease consultation prior to transplantation.

**Figure 2 pathogens-12-00074-f002:**
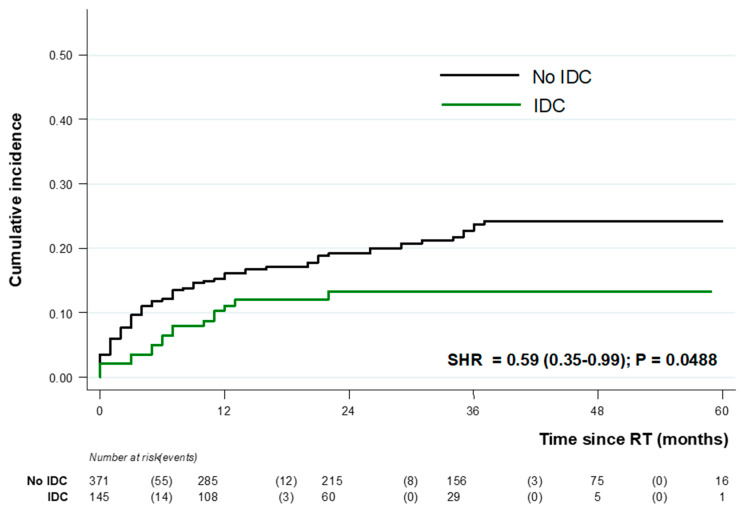
Cumulative incidence of respiratory tract infections (RTI) according to pretransplant infectious disease consultation: subhazard ratios (SHR) from Fine–Gray competing risks regression analysis.

**Figure 3 pathogens-12-00074-f003:**
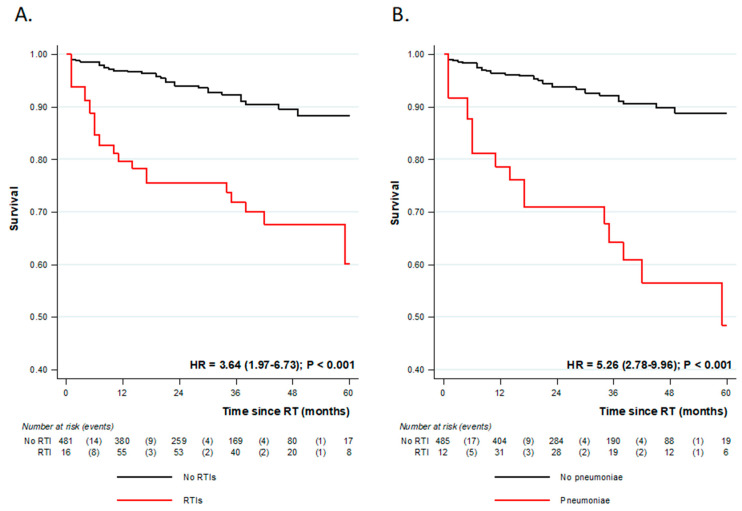
Overall survival curves according to the occurrence of (**A**) respiratory tract infection (RTIs) or (**B**) pneumoniae. Hazard ratios (HR) from time-dependent Cox regression analysis.

**Table 1 pathogens-12-00074-t001:** Baseline characteristics of the study population.

Variables	Whole Cohort	RTI Group	Control Group	*p*-Value
	*n* = 516	*n* = 95	*n* = 421	
**Recipients characteristics**				
Age, years, median (IQR)	55.6 (44.1–64.8)	59.0 (47.2–66.5)	55 (43.6–64.5)	**0.034**
Sex, Female, *n* (%)	190 (36.8)	36 (37.9)	154 (36.6)	0.81
Initial nephropathy				
Glomerulopathy, *n* (%)	130 (25.2)	21 (22.1)	109 (25.9)	0.44
Unknown, *n* (%)	120 (23.3)	19 (20)	101 (24)	0.41
Other, *n* (%)	41 (7.9)	17 (17.9)	24 (5.7)	**0.0002**
Diabete, *n* (%)	80 (15.5)	17 (17.9)	63 (15)	0.48
Hypertension, *n* (%)	41 (7.9)	11 (11.6)	30 (7.1)	0.15
Interstitial, *n* (%)	34 (6.6)	4 (4.2)	30 (7.1)	0.31
Genetic, *n* (%)	61 (11,8)	3 (3,2)	58 (13,8)	**0.008**
Obstructive, *n* (%)	9 (1.7)	3 (3.26)	6 (1.4)	0.26
Dialysis, *n* (%)	448 (86.8)	80 (84.2)	368 (87.4)	0.32
HIV+, *n* (%)	34 (6.6)	16 (16.8)	18 (4.3)	**<0.0001**
HCV+, *n* (%)	29 (5.6)	16 (16.7)	13 (3.1)	**<0.0001**
CMV+, *n* (%)	413 (80.4)	82 (86.3)	331 (79)	0.13
**Donors characteristics**				
Age, years, mean (SD)	58 (46.5–68)	62 (50–69)	57 (46–68)	0.12
Extended criteria donor, *n* (%)	262 (51)	56 (58.9)	206 (49.2)	0.09
Living donor, *n* (%)	77 (14.9)	10 (10.5)	67 (15.9)	0.19
CMV+, *n* (%)	325 (63.1)	67 (70.5)	258 (61.4)	0.1
**Sensitization risk factors**				
Former kidney transplantation, *n* (%)	59 (11.4)	13 (13.7)	46 (10.9)	0.45
Donor specific anti HLA antibodies, *n* (%)	188 (38.2)	44 (48.4)	144 (35.9)	**0.03**
**Kidney transplant characteristics**				
Cold ischemia time, hour, median (IQR)	16 (11.9–20.85)	15.83 (12.12–20)	16 (11.9–21)	0.81
**Immunosuppressive therapy**				
Induction, *n* (%)	509 (98.6)	95 (100)	414 (98.3)	0.21
Basiliximab, *n* (%)	172 (33.3)	32 (33.7)	140 (33.3)	0.94
Anti thymocyte globulin, *n* (%)	337 (65.3)	63 (66.3)	274 (65.1)	0.82
Rituximab, *n* (%)	45 (9)	10 (10.5)	35 (8.6)	0.56
Maintenance immunosuppressive therapy				
Calcineurin inhibitors, *n* (%)	471 (91.2)	85 (89.5)	386 (91.7)	0.6
Mycophenolate mofetil, *n* (%)	389 (75.7)	69 (72.6)	320 (76.4)	0.443
Belatacept, *n* (%)	40 (7.8)	11 (11.6)	29 (7)	0.137
Steroids, *n* (%)	511 (99.2)	95 (100)	416 (99)	0.34
**White blood cells before transplantation**				
Leucocytes (G/L), median (IQR)	6.4 (5.2–8.2)	6.6 (5.1–8.3)	6.2 (5.2–8.1)	0.867
Neutrophils (G/L), median (IQR)	4.1 (3.1–5.5)	4.45 (3.1–5.8)	4.1 (3.2–5.4)	0.561
Lymphocytes (G/L), median (IQR)	1.3 (1–1.7)	1.3 (0.9–1.7)	1.3 (1–1.7)	0.54

IQR: interquartile range; SD: standard deviation. HIV, Human Immundeficiency Virus; HCV, Hepatitis C Virus; CMV, Cytomegalovirus. Bolded results are statistically significant at the *p* < 0.05 level.

**Table 2 pathogens-12-00074-t002:** Follow-up of the patients included in the study.

Variables	Whole Cohort	RTI Group	Control Group	*p*-Value
	*n* = 516	*n* = 95	*n* = 421	
**Acute rejection, *n* (%)**	85 (16.5)	10 (10.5)	75 (17.8)	0.34
T cell mediated	55 (10.7)	7 (7.4)	48 (11.4)	0.21
Antibody-mediated	18 (3.5)	1 (1.1)	17 (4)	0.17
Mixed	12 (2.3)	2 (2.1)	10 (2.4)	0.79
Time from transplantation, months (median, IQR)	3.5 (2–12)	0 (0–2)	4 (2–12)	**0.01**
**Viral infections M12**				
BK viruria	78 (19.5)	14 (15.2)	64 (20.7)	0.24
BK viremia	23 (5.7)	10 (10.9)	13 (4.1)	**0.014**
CMV viremia	32 (7.9)	14 (15.2)	18 (5.8)	**0.003**
**12 month follow up**				
eGFR ml/min/1,73 m2	45.9 (34.1–59.0)	42.7 (31.9–57.9)	46.8 (35.1–59.5)	0.98
**Last follow-up**				
Allograft loss, *n* (%)	33 (6.4)	4 (4.2)	29 (6.9)	0.55
Death, *n* (%)	54 (10.5)	22 (23.2)	32 (7.6)	**<0.0001**
Time from transplantation, months (median, IQR)	12 (4–30)	10.5 (5–34)	14.5 (2–29)	0.9

IQR: interquartile range. CMV, Cytomegalovirus; eGFR, estimated Glomerular Filtration Rate. Bolded results are statistically significant at the *p* < 0.05 level.

**Table 3 pathogens-12-00074-t003:** Independent risk and protective factors of respiratory tract infections: results from multivariate logistic regression.

Variables	ORa	95% CI	*p*-Value
Recipient age	1.02	1.00–1.04	**0.02**
Infectious disease consultation	0.48	0.26–0.88	**0.02**
HIV+	4.2	1.87–9.45	**0.0005**
HCV+	4.2	1.79–9.79	**0.001**
Donor specific anti-HLA antibodies before transplantation	1.68	1.03–2.73	**0.036**

ORa, adjusted odds ratio from logistic regression model; CI, Confidence interval; HIV, Human Immundeficiency Virus; HCV, Hepatitis C Virus; HLA, Human Leucocyte Antigen. Bolded results are statistically significant at the *p* < 0.05 level.

**Table 4 pathogens-12-00074-t004:** Independent risk factors of all-cause death: results from time dependent Cox proportional hazards regression.

Variables	HRa	95% CI	*p*-Value
** *Model 1* **			
Respiratory tract infection: *all kinds*	1.93	1.002–3.70	**0.049**
Recipient age	1.08	1.05–1.11	**<0.0001**
HIV+	3.7	1.63–8.42	**0.002**
HCV+	2.48	1.14–5.41	**0.02**
Combined kidney liver transplant	4.8	2.15–10.72	**0.0001**
** *Model 2* **			
Respiratory tract infection: *pneumoniae*	2.15	1.08–4.27	**0.03**
Recipient age	1.07	1.04–1.11	**<0.0001**
HIV+	3.76	1.68–8.42	**0.001**
HCV+	2.47	1.15–5.34	**0.021**
Combined kidney liver transplant	4.51	2.00–10.20	**0.0003**

HRa, adjusted hazard ratio from time dependent Cox model treating respiratory tract infection as a time varying covariate; CI, Confidence interval; HIV, Human Immundeficiency Virus; HCV, Hepatitis C. Bolded results are statistically significant at the *p* < 0.05 level.

## Data Availability

Data are available upon reasonable request to the corresponding author.

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
