# Peer review of "Impact of a Dedicated Pretransplant Infectious Disease Consultation on Respiratory Tract Infections in Kidney Allograft Recipients: A Retrospective Study of 516 Recipients"

_pathogens, 2023, doi:10.3390/pathogens12010074_

Round 1
Reviewer 1 Report
This is a well documented study.
Comments:
1. What is the connection between RTIs and the rest of the analysed conditions (HCV, HIV, age, donor-specific anti-HLA antibodies, etc)? In case they were studied for reasons of comparison to the RTI data or as factors involved with the RTIs, this should be mentioned, otherwise the inclusion of unrelated clinical conditions is rather confusing and inconsistent with the paper title.
2. A few typing errors were noticed, such as: L. 251, “are frequent complications” instead of “are a frequent complications”.
Author Response
This is a well-documented study.
Comments:
- What is the connection between RTIs and the rest of the analysed conditions (HCV, HIV, age, donor-specific anti-HLA antibodies, etc)? In case they were studied for reasons of comparison to the RTI data or as factors involved with the RTIs, this should be mentioned, otherwise the inclusion of unrelated clinical conditions is rather confusing and inconsistent with the paper title.
We thank the reviewer for the comment. These variables are consistently reported and exploited in studies investigating such patient populations and are known to influence the immune state and susceptibilities to infection.
- A few typing errors were noticed, such as: L. 251, “are frequent complications” instead of “are a frequent complications”.
We modified accordingly
Reviewer 2 Report
In this retrospective study the AA report the impact of pretransplant infectious disease consultation (IDC) in 519 patients of whom 145 had ICD before tx. They conclude that patients who had benefited of IDC presented significantly less respiratory tract infections RTIs (p = 0.049). RTIs were an independent risk factor of mortality (HR = 3.64). Independent risk factors for RTIs included HIV (OR = 3.33) and HCV (OR = 3.76). RTIs after kidney transplant are an independent risk factor of death. IDC was identified as an independent protective factor No original conclusion from the univariate and multivariate analysis. The novelty of the paper is that IDC was found to be an independent protective factor for infection in a competing risk time-to-event analysis framework that, as the AA acknowledge, needs to be confirmed with more studies. The paper is well written. The incidence and type of infection definitely reflects the changed demographic and clinical characteristic of patients eligible for kidney transplantation. Unfortunately, there are some missing points: · A descriptive part explaining the suggestion after IDC: ie: Were all the patients Caucasian? Was the immunosuppressive regimen modified in the high-risk populations? Was some prophylaxis added? Was hospitalization reduced if possible? In the discussion are briefly mentioned the vaccination suggested prior to transplantation. · Patients who survived the infection, had a change of immunosuppression, were strictly followed-up? · Why not all the HIV/HCV or combined liver-kidney tx patients underwent IDC? · On what base the IDC was requested? Just to update recommended vaccinations (page 3, line 102)? · The time interval of the study is 2015-2019, surprisingly m-Tor inhibitors are not present among the immunosuppressive schedules: Center protocol? · We understand that the aim of the paper is to report on the impact of IDC before tx but an analysis including other variables such as correlation between immunosuppressive schedule and RTIs would have been useful. In the present form the paper cannot be published: it does not give suggestions applicable exstensively.Author Response
In this retrospective study the AA report the impact of pretransplant infectious disease consultation (IDC) in 519 patients of whom 145 had ICD before tx. They conclude that patients who had benefited of IDC presented significantly less respiratory tract infections RTIs (p = 0.049). RTIs were an independent risk factor of mortality (HR = 3.64). Independent risk factors for RTIs included HIV (OR = 3.33) and HCV (OR = 3.76). RTIs after kidney transplant are an independent risk factor of death. IDC was identified as an independent protective factor No original conclusion from the univariate and multivariate analysis.
The novelty of the paper is that IDC was found to be an independent protective factor for infection in a competing risk time-to-event analysis framework that, as the AA acknowledge, needs to be confirmed with more studies. The paper is well written. The incidence and type of infection definitely reflects the changed demographic and clinical characteristic of patients eligible for kidney transplantation.
We thank the reviewer for highlighting the novelty of our findings.
Unfortunately, there are some missing points: ·
A descriptive part explaining the suggestion after IDC: ie: Were all the patients Caucasian?
We thank the reviewer for the comment.
All patients that underwent kidney transplant were included, independently of ethnicity. Therefore, patients were from diverse geographical origins, however French ethical legislation forbid us reporting race and ethnicity characteristics unless justified. We nevertheless noted in the methods section a that all patients were included, implying that ethnicity did not influence our inclusion criteria.
Was the immunosuppressive regimen modified in the high-risk populations?
Immunosuppressive regimen was decided following local protocol by the nephrologist and depending on rejection risk score and personal history. Importantly however, we did not note differences in immunosuppressive regimen in between patient that developed respiratory tract infections and patients that did not.
Was some prophylaxis added?
The management for CMV prophylaxis followed international recommendations. Prophylaxis involved the administration of oral valganciclovir to high (D+/R-) and intermediate (R+ treated with thymoglobulin) risk patients.
Duration of prophylaxis was 6 months in high-risk patients and 3 months in intermediate ones. Participants with past history of tuberculosis were treated with isoniazid for 6 months after transplantation.
Pneumocystis jirovecii prophylaxis included trimethoprim-sulfamethoxazole (400 mg per day) or pentacarinat aerosol for 12 months after transplantation and untill CD4 count were above 200/mm3.
We accordingly modified the methods section to reference these recommendations.
Was hospitalization reduced if possible?
Immunocompromised patients with pneumonia were almost systematically hospitalized according to standard criteria for hospitalization, whereas patients with bronchitis were hospitalized according to clinical severity criteria. Since we noted significantly less respiratory tract infections in patients that benefited IDC, we can reasonably assume less hospitalization. Nevertheless, this data was not collected and we cannot confidently assert this conclusion. Data regarding delay to hospital discharge was not collected either.
In the discussion are briefly mentioned the vaccination suggested prior to transplantation. Patients who survived the infection, had a change of immunosuppression, were strictly followed-up? ·
Data regarding the change of immunosuppression regimen after infection was not collected. Frequency of consultation post-infection was determined by the transplantation nephrologist and at the physician discretion.
Why not all the HIV/HCV or combined liver-kidney tx patients underwent IDC? ·
We thank the reviewer for this comment. IDC is a dedicated consultation prior to transplantation. It is performed regardless of other indications that also require an infectious disease expertise such as HIV or HCV infection. IDC was set up progressively from 2014 and patients benefited IDC independently of their risk factors or their prior exposure to infections.
On what base the IDC was requested? Just to update recommended vaccinations (page 3, line 102)? ·
We thank the reviewer for the comment. IDC prior transplantation has several aims including vaccine update, tuberculosis tracking, diagnosis and eventual treatment and an evaluation of personal risk factors. In more details:
- Vaccinations
Informations regarding types and timing of previous vaccinations against hepatitis A and B, Pneumococcus, Influenza, tetanus-diphteria-polio-acellular pertussis (TDPaP) were recorded. Serological status of hepatitis A and B, varicella-zoster virus (VZV), measles and yellow fever in previously vaccinated patients were evaluated.
The catch up strategy for vaccinations based on French national vaccines schedule and recommendations for immunocompromised patients/patients awaiting for solid organ transplantation was proposed and personalized as follow:
- All patients: Flu and Pneumococcus vaccinations
- Only if a recall dose needed: tetanus-diphteria-polio-acellular pertussis (TDPaP) vaccination
- Older than 65 years without previous immunosuppression: attenuated VZV vaccine
- Younger than 19 years, female or men having sex with men (MSM): papillomavirus vaccine
- On demand if need (travel or catch-up doses): measles, mumps, rubella (MMR), VZV, meningococcus, haemophilus influenza, yellow fever hepatitis A and hepatitis B vaccinations. Yellow fever is mandatory for travelers to endemic intertropical regions of Africa and South America.
- Tuberculosis
A full clinical history survey addressing past tuberculosis (TB) episodes or TB exposure was performed. The pretransplant chest X-ray and/or computed tomography (CT) scan were reviewed in all patients to look for active TB lesions or old healed TB lesions in asymptomatic patient.
All patients included in the study underwent LTBI screening using Quantiferon (QTF, Qiagen GmbH, Germany), an Interferon gamma release assays (IGRA). LTBI treatment was proposed to all quantiferon positive patients and/or patients with radiological old healed TB lesions, not previously treated for tuberculosis, as follow: isoniazid alone at 3mg/kg dosing for 9 months or combination of rifampicin (10mg/kg) and isoniazid (3mg/kg).
- Evaluation of individual risk factors
Clinical (past personal history), biological and radiological evaluation was performed to evaluate individual infectious and post-transplant prophylaxis was adapted following recommendations.
These details have been resumed in the manuscript (in the methods section).
The time interval of the study is 2015-2019, surprisingly m-Tor inhibitors are not present among the immunosuppressive schedules: Center protocol? ·
Immunosuppressive regimen is set up according to local protocols and do not include mTOR inhibitors at the induction or the maintenance phase. Immunosuppressive regimen can be modified by the nephrologist during follow-up but this data was unfortunately not collected.
We understand that the aim of the paper is to report on the impact of IDC before tx but an analysis including other variables such as correlation between immunosuppressive schedule and RTIs would have been useful.
Unfortunately, we did not collect data regarding immunosuppressive schedule and modifications during follow-up. Importantly however, we found that immunosuppressive regimen at induction and following transplantation did not significantly influence the occurrence of RTIs.
Round 2
Reviewer 2 Report
In the corrected version of the manuscript I can only find very little of the modification requested. The point of a review is implementing and ameliorating the paper. Any comment unless self explaining, need to be inlcuded in the reviewed version ie:
-Immunosuppressive regimen was decided following local protocol by the nephrologist and depending on rejection risk score and personal history. Importantly however, we did not note differences in immunosuppressive regimen in between patient that developed respiratory tract infections and patients that did not.....
The paper needs to be reviewed
Author Response
We thank the reviewer for the comment. Please see the attachment.

Round 3
Reviewer 2 Report
The corrections made have improved the paper